# A Comparison of the Photophysical, Electrochemical and Cytotoxic Properties of *meso*-(2-, 3- and 4-Pyridyl)-BODIPYs and Their Derivatives

**DOI:** 10.3390/s22145121

**Published:** 2022-07-07

**Authors:** Caroline Ndung’u, Daniel J. LaMaster, Simran Dhingra, Nathan H. Mitchell, Petia Bobadova-Parvanova, Frank R. Fronczek, Noémie Elgrishi, Maria da Graça H. Vicente

**Affiliations:** 1Department of Chemistry, Louisiana State University, Baton Rouge, LA 70803, USA; cndung2@lsu.edu (C.N.); daniel.j.lamaster@gmail.com (D.J.L.); sdhing1@lsu.edu (S.D.); nmitc23@lsu.edu (N.H.M.); ffroncz@lsu.edu (F.R.F.); noemie@lsu.edu (N.E.); 2Department of Chemistry and Fermentation Sciences, Appalachian State University, Boone, NC 28608, USA; bobadovap@appstate.edu

**Keywords:** BODIPY, pyridinium, fluorescence

## Abstract

Boron dipyrromethene (BODIPY) dyes bearing a pyridyl moiety have been used as metal ion sensors, pH sensors, fluorescence probes, and as sensitizers for phototherapy. A comparative study of the properties of the three structural isomers of *meso*-pyridyl-BODIPYs, their 2,6-dichloro derivatives, and their corresponding methylated cationic pyridinium-BODIPYs was conducted using spectroscopic and electrochemical methods, X-ray analyses, and TD-DFT calculations. Among the neutral derivatives, the **3Py** and **4Py** isomers showed the highest relative fluorescence quantum yields in organic solvents, which were further enhanced 2-4-fold via the introduction of two chlorines at the 2,6-positions. Among the cationic derivatives, the **2catPy** showed the highest relative fluorescence quantum yield in organic solvents, which was further enhanced by the use of a bulky counter anion (PF_6_^−^). In water, the quantum yields were greatly reduced for all three isomers but were shown to be enhanced upon introduction of 2,6-dichloro groups. Our results indicate that 2,6-dichloro-*meso*-(2- and 3-pyridinium)-BODIPYs are the most promising for sensing applications. Furthermore, all pyridinium BODIPYs are highly water-soluble and display low cytotoxicity towards human HEp2 cells.

## 1. Introduction

Since the synthesis of the first boron dipyrromethene (BODIPY) in the 1960s by Treibs and Kreuzer [1], BODIPY-based dyes have attracted great attention [2,3,4,5]. This is due to their desirable properties which include large molar absorption coefficients, high photo and chemical stability [2], tunable photophysical properties [3,4], narrow spectral band width, and high fluorescence quantum yields in organic solutions [5]. BODIPYs have been developed for a wide range of applications, including photosensitizers in cancer therapy [6,7], biological labels [8,9,10,11], imaging agents [12], fluorescent switches [13,14,15], and laser dyes [16,17].

While there has been a large number of BODIPY dyes designed, synthesized and characterized over the last two decades, the development and synthesis of low molecular weight and water-soluble BODIPY dyes is yet to be fully explored [18]. Additional studies in this area are needed to provide water-soluble, membrane-permeable BODIPY derivatives and widen their biological applications [19]. The water solubility of BODIPY dyes has been enhanced via installation of solubilizing groups such as phosphates [20,21], sulfonates [22,23,24,25], carboxylates [26,27,28], carbohydrates [29,30], and oligoethylene glycol chains [31]. However, these solubilizing groups increase the dye’s molecular size and tend to decrease their cell membrane permeability [32]. On the other hand, pyridinium groups are easy to install, and lead to cationic BODIPY derivatives with enhanced water-solubility, stability, and membrane permeability. In addition, *meso*-pyridyl-BODIPYs have been used as ligands in coordination chemistry, and as fluorescent pH sensors; therefore, this type of functionalized BODIPYs has found applications in bioimaging and biosensing. However, previous studies on a *meso*-(4-pyridinium)-BODIPY showed that upon excitation, the dye undergoes donor-photoinduced electron transfer (d-PeT) to a charge-transfer state that is quenched through intersystem crossing (ISC) [32]. This process occurs by the excitation (S_0_→S_2_) and subsequent non-radiative decay from (S_2_→S_1_) moving the electron from HOMO→LUMO+1→LUMO as shown in Figure 1. In this charge-transfer process, the 4-pyridinium group acts as an acceptor and the singlet-excited state of the BODIPY core acts as the donor. As a result, the fluorescence quantum yield of the dye is greatly reduced (see Figure 2a). While restoration of the *meso*-(4-pyridyl)-BODIPY fluorescence can be achieved by either inhibiting d-PeT or the ISC, the charge recombination emission can be enhanced to varying degrees by increasing the electrostatic charge of the ‘hole’ created by the electronic excitation and subsequent d-PeT process. We recently showed [33] that this can be accomplished through the installation of electron-withdrawing groups at the 2,6-positions. As an example, chlorination at the 2,6-positions induced an increase in the fluorescence quantum yield of a *meso*-4-pyridinium BODIPY from 0.019 to 0.048 in acetonitrile and from 0.004 to 0.038 in water [33]. Furthermore, Ueno and coworkers [34] reported that the installation of acetyl groups at the 2,6-positions of a *meso*-(nitroanisole)-BODIPY increased the fluorescence quantum yield of the dye in acetonitrile by over 20-fold (see Figure 2b).

Herein we report the synthesis and investigation of *meso*-(2-pyridyl) and *meso*-(3-pyridyl)-BODIPYs, and of their 2,6-dichloro and pyridinium derivatives (see Figure 3) and compare their photophysical, electrochemical and cytotoxic properties relative to the known *meso*-(4-pyridyl) and *meso*-(4-pyridinium) derivatives and their 2,6-dichloro analogs. The moderate electron-withdrawing chlorine atoms are easily introduced at the 2,6-positions and expected to induce red-shifts in the absorption and emission wavelengths, increase reduction potentials, increase stability, and enhance the relative fluorescence quantum yields in both organic solvents and water. This type of BODIPY and halogenated derivatives could find multiple applications as optical sensors, fluorescent probes, or as sensitizers for phototherapy.

## 2. Materials and Methods

### 2.1. Synthesis and Characterization

#### 2.1.1. General

Commercially available reagents and solvents were used as received from VWR or Sigma Aldrich unless noted otherwise. All reactions were monitored by thin-layer chromatography (TLC) using 0.2 mm silica gel plates (with UV indicator, polyester backed, 60 Å, pre-coated). Liquid chromatography was performed on preparative TLC plates or via silica gel column chromatography (60 Å, 230–400 mesh). NMR spectra were measured on 400 or 500 MHz for ^1^H and 500 or 750 MH for ^13^C spectrometer. Chemical shifts (δ) are given in parts per million (ppm) In CDCl_3_ (7.27 ppm for ^1^H NMR, 77.0 ppm for ^13^C NMR) or (CD_3_)_2_CO (2.05 ppm for ^1^H NMR, 206.68 and 29.92 ppm for ^13^C NMR) or CD_3_CN (1.94(5) ppm for ^1^H NMR, 118.69 and 1.39 ppm for ^13^C NMR); coupling constants (J) are given in hertz. High-resolution mass spectra (HRMS) were obtained using an Agilent 6230-B ESI-TOF mass spectrometer.

BODIPYs **2Py** [35] and **3Py** [36] were prepared following a procedure previously reported, by reacting 2,4-dimethylpyrrole with the corresponding pyridine 2-carbaldehyde in the presence of trifluoroacetic acid (TFA), followed by oxidation with 2,3-dichloro-5,6-dicyano-1,4-benzoquinone (DDQ) and complexation with BF_3_•(OEt)_2_ in triethylamine. The spectroscopic data for **2Py** and **3Py** are in agreement with the literature reports.

#### 2.1.2. 1,3,5,7-Tetramethyl-8-(N-methyl-2-pyridyl)-BODIPY Iodide (**2catPy**)

BODIPY **2Py** (13.31 mg, 0.041 mmol) was dissolved in dry acetonitrile (3 mL) in an oven dried 10 mL round bottom flask. An excess of methyl iodide (5 mL, 80.32 mmol) was added, and the mixture was refluxed for 8 h under N_2_ atmosphere. The solvent was removed under reduced pressure. The grayish residue was washed with petroleum ether (5 mL) and the resulting solid recovered via filtration to afford 13 mg, 98% yield of **2catPy**. ^1^H NMR (500 MHz, CD_3_CN) δ 9.06 (d, *J* = 6.2 Hz, 1H), 8.70 (t, *J* = 7.5 Hz, 1H), 8.27 (m, *J* = 16.8, 7.3 Hz, 2H), 6.25 (s, 2H), 4.24 (s, 3H), 2.55 (s, 6H), 2.15 (s, 6H). ^13^C NMR (126 MHz, CD_3_CN) δ 161.37, 149.76, 148.96, 143.77, 132.24, 130.86, 124.38, 47.75, 15.53, 13.92. ^11^B NMR (128 MHz, acetone-d_6_) δ 0.57 (t, *J* = 31.5 Hz). HRMS (ESI-TOF) m/z [M]^+^ calcd for C_19_H_21_BF_2_N_3_: 339.1827 found 339.1830.

#### 2.1.3. 1,3,5,7-Tetramethyl-8-(N-methyl-3-pyridyl)-BODIPY Iodide (**3catPy**)

This compound was prepared as described above for **2catPy**, using **3Py** (13.3 mg, 0.041 mmol), dry acetonitrile (3 mL) and methyl iodide (5 mL, 80.32 mmol). The title BODIPY was obtained in 98% yield as a grayish powder. ^1^H NMR (500 MHz, CD_3_CN) δ 8.90–8.86 (m, 2H), 8.62 (d, *J* = 8.0 Hz, 1H), 8.23–8.19 (m, 1H), 6.19 (s, 2H), 4.41 (s, 3H), 2.52 (s, 6H), 1.43 (s, 6H). ^13^C NMR (126 MHz, CD_3_CN) δ 159.29, 147.69, 144.62, 136.20, 130.20, 123.94, 50.41, 16.42, 15.30. ^11^B NMR (128 MHz, acetone-d_6_) δ 0.60 (t, *J* = 32.0 Hz). HRMS (ESI-TOF) m/z [M]^+^ calcd for C_19_H_21_BF_2_N_3_: 339.1827 found 339.1828.

#### 2.1.4. 2,6-Dichloro-8-(2-pyridyl)-1,3,5,7-tetramethyl-BODIPY (**2PyCl_2_**)

A solution of BODIPY **2Py** (10.7 mg, 0.03 mmol) in dry dichloromethane (6 mL) was stirred at room temperature in an oven dried round bottom flask under N_2_ atmosphere. Trichloroisocyanuric acid (TCCA) (6.10 mg, 0.026 mmol) was dissolved in dichloromethane (2 mL) and added slowly to the solution. The reaction was stirred at room temperature for 2 h, until complete disappearance of the starting material. The solvent was removed under reduced pressure and the resulting residue was purified via column chromatography using 5–20% ethyl acetate in hexanes, to give 9 mg (84%) of the title compound. ^1^H NMR (500 MHz, CDCl_3_) δ 8.84 (d, *J* = 5.0 Hz, 1H), 7.91 (t, *J* = 7.7 Hz, 1H), 7.54–7.50 (m, 1H), 7.45 (d, *J* = 7.7 Hz, 1H), 2.61 (s, 6H), 1.32 (s, 6H). ^13^C NMR (126 MHz, CDCl_3_) δ 153.40, 153.14, 150.49, 139.18, 137.43, 137.23, 129.58, 124.39, 124.30, 29.72, 12.53, 11.36. ^11^B NMR (128 MHz, CDCl_3_) δ 0.43 (t, *J* = 31.7 Hz). HRMS (ESI-TOF) m/z [M+H]^+^ calcd for C_18_H_16_BCl_2_F_2_N_3_: 394.0858 found 394.0857.

#### 2.1.5. 2,6-Dichloro-8-(3-pyridyl)-1,3,5,7-tetramethyl-BODIPY (**3PyCl_2_**)

This compound was prepared as described above for **2PyCl_2_**, using BODIPY **3Py** (12.1 mg, 0.04 mmol) and TCCA (7.70 mg, 0.03 mmol). The reaction was purified via column chromatography using 5–20% ethyl acetate in hexanes yielding 10.8 mg (89%) of a pinkish product^. 1^H NMR (400 MHz, acetone-d_6_) δ 8.85 (d, *J* = 3.2 Hz, 1H), 8.69 (s, 1H), 7.95 (d, *J* = 7.8 Hz, 1H), 7.70–7.64 (m, 1H), 2.55 (s, 6H), 1.40 (s, 6H).^13^C NMR (126 MHz, acetone-d_6_) δ 152.66, 150.86, 148.20, 139.81, 137.89, 136.13, 130.20, 129.78, 124.11, 11.75. ^11^B NMR (128 MHz, CDCl_3_) δ 0.55, 0.30, 0.06. HRMS (ESI-TOF) m/z [M+H]^+^ calcd for C_18_H_16_BCl_2_F_2_N_3_: 394.0858 found 394.0870.

#### 2.1.6. 2,6-Dichloro-8-(N-methyl-2-pyridyl)-1,3,5,7-tetramethyl-BODIPY (**2catPyCl_2_**)

This compound was prepared as described above for **2catPy**, using BODIPY **2PyCl_2_** (15 mg, 0.038 mmol), anhydrous acetonitrile (10 mL), and methyl iodide (5 mL). The title compound was obtained (13.4 mg, 89% yield). ^1^H NMR (400 MHz, DMSO-d_6_) δ8.88 (d, *J* = 6.3 Hz, 1H), 8.41 (t, *J* = 7.9 Hz, 1H), 8.12 (d, *J* = 7.9 Hz, 1H), 8.02 (t, *J* = 7.1 Hz, 1H), 3.82 (s, 3H), 2.12 (s, 6H), 0.88 (s, 6H). ^13^C NMR (176 MHz, CD_3_CN) δ 158.12, 150.14, 149.25, 138.47, 132.29, 131.30, 129.70, 47.96, 13.65, 11.81. ^11^B NMR (128 MHz, acetone-d_6_) δ 0.50, 0.26, 0.02. HRMS (ESI-TOF) m/z [M]^+^ calcd for C_19_H_19_BCl_2_F_2_N_3_: 408.1015 found 408.1011.

#### 2.1.7. 2,6-Dichlo-8-(N-methyl-3-pyridyl)-1,3,5,7-tetramethyl-BODIPY (**3catPyCl_2_**)

This compound was prepared as described above for **3catPy**, using BODIPY **3PyCl_2_** (10 mg, 0.025 mmol), anhydrous acetonitrile (10 mL), and methyl iodide (4 mL). The title compound was obtained (8.6 mg, 86% yield). ^1^H NMR (400 MHz, CD_3_CN) δ 8.94 (d, *J* = 6.1 Hz, 1H), 8.90 (s, 1H), 8.62 (d, *J* = 8.1 Hz, 1H), 8.28–8.23 (m, 1H), 4.43 (s, 3H), 2.57 (s, 6H), 1.44 (s, 6H). ^13^C NMR (126 MHz, CD_3_CN) δ 156.11, 148.72, 147.60, 145.72, 139.37, 135.25, 130.41, 50.60, 14.19. ^11^B NMR (128 MHz, Acetone) δ 0.55, 0.31, 0.07. HRMS (ESI-TOF) m/z [M]^+^ calcd for C_19_H_19_BCl_2_F_2_N_3_: 408.1015 found 408.1017.

#### 2.1.8. Anion Exchange

Anion exchange was accomplished by treating the corresponding iodide cationic BODIPY (1 equivalent) in acetonitrile with excess NH_4_PF_6_ in methanol at room temperature. The mixture allowed to sit for 1 h and solvent evaporated under reduced pressure. The solid was then washed with water and the PF_6_ salt recovered via filtration. HRMS (ESI-TOF) m/z [M+H]^+^ calcd for C_19_H_21_BF_2_N_3_: 340.1794 found 340.1795 for **2catPy.** HRMS (ESI-TOF) m/z [M+H]^+^ calcd for C_19_H_21_BF_2_N_3_: 340.1794 found 340.1795 for **3catPy.** HRMS (ESI-TOF) m/z [M]^+^ calcd for C_19_H_21_BF_2_N_3_: 339.1827 found 339.1827 for **4catPy**. HRMS (ESI-TOF) m/z [M]^+^ calcd for C_19_H_19_BCl_2_F_2_N_3_: 408.1015 found 408.1017 for **2catPyCl_2_**. HRMS (ESI-TOF) m/z [M]^+^ calcd for C_19_H_19_BCl_2_F_2_N_3_: 408.1015 found 408.1015 for **3catPyCl_2_**. HRMS (ESI-TOF) m/z [M]^+^ calcd for C_19_H_19_BCl_2_F_2_N_3_: 408.1015 found 408.1015 for **4catPyCl_2_**.

### 2.2. Spectroscopy Methods

UV−vis absorption spectra were collected on a Varian Cary 50 Bio spectrophotometer. Emission spectra were obtained on a PerkinElmer LS55 spectrophotometer, at room temperature. Spectrophotometric grade solvents and quartz cuvettes (1 cm path length) were used. Relative fluorescence quantum yields (Φ_f_) were calculated using rhodamine 6G (Φ_f_ = 0.86 in methanol) as reference for the neutral BODIPYs and Ru(bpy)Cl_2_.6H_2_O (Φ_f_ = 0.028 in water) as the standard for the cationic BODIPYs using the following equation: Φ_x_ = Φ_st_ × Grad_x_/Grad_st_ × (η_x_/η_st_)^2^, where the Φ_X_ and Φ_ST_ are the quantum yields of the sample and standard, Grad_X_ and Grad_ST_ are the gradients from the plot of integrated fluorescence intensity vs. absorbance, and η represents the refractive index of the solvent (x is for the sample and st for the standard).

### 2.3. Electrochemistry

Cyclic voltammograms were collected on a Biologic SP-300 potentiostat using a three-electrode setup with a platinum working electrode (2 mm diameter, CH Instruments), platinum counter electrode (2 mm diameter, CH Instruments), and a silver wire pseudo reference-electrode stored in 0.25 M NBu_4_PF_6_ in anhydrous acetonitrile and separated from the electrochemical cell by a glass frit. The electrochemical cell was composed of a borosilicate glass scintillation vial with a custom-machined cap. Working electrodes were polished and then pre-treated by cycling between upper and lower boundaries of the electrochemical window for the experiment in a solution containing only the electrolyte. Measurements were taken in a solution of 0.25 M recrystallized NBu_4_PF_6_ in dried and degassed anhydrous acetonitrile. BODIPYs were added to 5 mL of electrolyte solution to give a concentration of 1 mM. Each solution also contained 1 mM ferrocene as an internal reference. Solutions were sparged with N_2_ before each measurement.

### 2.4. X-ray Crystallography

The crystal structures of **2Py**, **2PyCl_2_** and **3Py** were determined at T = 90 K from data collected on a Bruker Kappa Apex-II diffractometer equipped with Mo Kα (for **2Py** and **2PyCl_2_**) or CuKα (for **3Py**) source. Absorption corrections were by the multi-scan method. Non-hydrogen atoms were treated with anisotropic displacement parameters. The H atoms were visible in difference maps and were placed in idealized positions in the refinements, with torsional parameters refined for methyl groups. For **2Py** and **2PyCl_2_**, disorder exists in which the pyridyl group is rotated 180° about the bond joining it to the BODIPY core. Crystal Data: **2Py**, C_18_H_18_BF_2_N_3_, FW = 325.16, orthorhombic space group Pbca, a = 12.7201(8), b = 12.2790(8), c = 20.1538(13) Å, V = 3147.8(3) Å^3^, Z = 8, θ_max_ = 38.6°, R = 0.054 for the 5941 I > 2σ(I) data, wR(F^2^) = 0.146 for all 8744 data and 222 parameters. The CIF has been deposited at the Cambridge Crystallographic Data Centre, CCDC 2163983; **2PyCl2**, C_18_H_16_BCl_2_F_2_N_3_, FW = 394.05, monoclinic space group P2_1_/c, a = 6.4405(8), b = 15.6937(19), c = 17.205(2) Å, β = 93.150(4)°, V = 1736.4(4)Å^3^, Z = 4, θ_max_ = 29.2°, R = 0.067 for the 3122 I > 2σ(I) data, wR(F^2^) = 0.126 for all 4694 data and 240 parameters, CCDC 2163984; **3Py**, C_18_H_18_BF_2_N_3_, FW = 325.16, monoclinic space group P2_1_, a = 7.0759(2), b = 12.0387(3), c = 19.0447(5) Å, β = 96.182(2)°, V = 1612.88(7)Å^3^, Z = 4, θ_max_ = 69.5°, R = 0.030 for the 5618 I > 2σ(I) data, wR(F^2^) = 0.073 for all 6017 data and 441 parameters, Flack x = −0.03(5), CCDC 2163985.

### 2.5. Computational Methods

The geometries of the ground and excited states of all compounds were optimized without symmetry constraints at the cam-b3lyp/6-31+G(d,p) level in vacuum. This method has been shown to correctly reproduce the experimental trends [18]. The UV-vis absorption data were calculated using the TD-DFT method at the same level. The rotational barriers were calculated in CH_3_CN taking the solvent effects into account using the Polarized Continuum Model (PCM). All calculations were performed using the Gaussian 09 program package [37].

### 2.6. Cell Toxicity

Human carcinoma HEp2 cells were obtained from ATCC and maintained in a 75 cm^2^ flask (Chemglass) with 1× MEM (Eagle’s minimal essential medium) containing 10% FBS (fetal bovine serum) and 1% antibiotic (penicillin–streptomycin) in a humidified, 5% CO_2_ incubator at 37 °C. Stock solutions with a concentration of 32 mM in DMSO were prepared for each BODIPY. The stock solution was diluted to the final working concentrations with culture medium. The HEp2 cells were plated at 7500–10,000 cells per well in a Costar 96-well plate and allowed to grow for 24 h. The BODIPY stock solution was diluted to a 200 μM working solution and two-fold serial dilutions were prepared with final concentrations of 200, 100, 50, 25, 12.5, 6.25, and 0 μM. Each BODIPY was incubated for 24 h at 37 °C under 5% CO_2_. For the dark cytotoxicity experiments, after incubation, excess BODIPY was removed by washing the cells three times with 1× PBS (phosphate-buffered saline) and replacing with media containing 20% Cell Titer Blue (Promega, Madison, WI, USA). The cells were then incubated for additional 24 h at 37 °C under 5% CO_2_ and the cell toxicity was determined by fluorescence intensity at ex 570/em 615 nm using a FluoStar Optima microplate reader (BMG Labtech, Cary, NC, USA).

For the phototoxicity experiments, after incubation the cells were exposed to a light dose of ~1.5 J/cm^2^ generated using a 600 W Quartz Tungsten Halogen lamp (Newport Corporation, Irvine, CA, USA). The cells were exposed for 20 min while the 96-well plate rested on an EchoTherm chilling plate (Torrey Pines Scientific, Carlsbad, CA, USA) set to 5 °C. Following light exposure, the loading media was removed and media containing 20% Cell Titer Blue was added. After incubation for 24 h at 37 °C under 5% CO_2_, the cell viability was determined as described above.

## 3. Results and Discussion

### 3.1. Synthesis

The **2Py** [35] and **3Py** [36] BODIPYs were prepared in 48% and 55% yields, respectively, by condensation of 2,4-dimethylpyrrole with the corresponding pyridine carboxaldehyde in presence of TFA, followed by oxidation with DDQ and boron complexation using boron trifluoride etherate, as previously reported. Treating BODIPYs **2Py** and **3Py** with excess of methyl iodide in acetonitrile gave the corresponding pyridinium BODIPYs, **2catPy** and **3catPy**, in nearly quantitative yields. Anion exchange was performed on these BODIPYs using ammonium hexafluorophosphate (NH_4_PF_6_). The resulting salts were purified by washing with organic solvents and characterized by mass spectrometry.

Chlorination of BODIPYs **2Py** and **3Py** using TCCA in dichloromethane [33] produced the corresponding 2,6-dichloro derivatives **2PyCl_2_** and **3PyCl_2_** in 84% and 89% yields, respectively. Methylation of these 2,6-dichloro-BODIPYs using methyl iodide afforded their pyridinium derivatives **2catPyCl_2_** and **3catPyCl_2_**. Anion exchange was performed on these BODIPYs using ammonium hexafluorophosphate. The structures of all the BODIPYs were confirmed by ^1^H, ^13^C and ^11^B NMR (see Appendix A, Appendix A) and by high-resolution mass spectrometry (ESI-TOF).

### 3.2. X-ray Analysis

Crystals of **2Py**, **3Py** and **2PyCl_2_** were obtained by slow evaporation of dichloromethane and pentane, and the results are shown in Figure 4. BODIPYs **2Py**, **3Py**, and **2PyCl_2_** all have their 12-atom BODIPY cores nearly planar, with mean deviations from coplanarity of 0.029, 0.023 and 0.038 Å, respectively. The dihedral angles between the BODIPY core planes and the pyridyl planes are also similar, 79.3° for **2Py**, 82.3° for **3Py** (average of two independent molecules), and 85.9° for **2PyCl_2_**. These values agree well with those for the published structure of **4Py [36,38,39]**, for which the mean deviation from the BODIPY core is 0.018 Å and the BODIPY/Py dihedral angle is 83.9° (average of two). Bond distances and angles also show excellent agreement across the three structures and in agreement with the published structure of **3Py** [36]. The relatively lower dihedral angle observed for **2Py** relative to the **3Py** and **4Py** suggests a slight increased flexibility for **2Py**, although the difference is very small. Therefore, for all three isomers the pyridine is interlocked between the 1,7-methyl substituents and does not affect the molecular structure of the BODIPY core.

### 3.3. Spectroscopic Properties

The absorption and emission spectra of BODIPYs **2Py, 3Py, 2PyCl_2_, 3PyCl_2_** and their cationic derivatives **2catPy, 3catPy, 2catPyCl_2_, 3catPyCl_2_** were obtained at room temperature in acetonitrile, methanol, and water, and the results are as summarized in Table 1, and in Appendix A and Appendix A of the Appendix A. As expected, the absorption spectra for **2Py, 3Py** and **4Py** featured a sharp peak at ca. 501 nm which is attributed to the BODIPYs strong S_0_-S_1_ (π-π*) transition, as confirmed by molecular modeling. The broad high energy shoulder bands are attributed to the BODIPYs vibrational transitions. The emission peak for these BODIPYs is seen at ca. 514 nm, and the relatively small Stokes shift indicates that the molecular structure of the excited state is not very different than that of the ground state. The maximum absorption and emission wavelengths for the neutral **2Py, 3Py** and **4Py** are almost identical in both acetonitrile and methanol (see Table 1 and Appendix A). In agreement with this observation, computational modeling of the absorption spectra in the gas phase confirms the almost identical maximum absorption wavelengths, which correspond well to the almost identical HOMO-LUMO gaps, as can be seen in Table 2. For the entire series of **2Py, 3Py** and **4Py**, the HOMO is almost entirely localized on the BODIPY core, whereas the LUMO partially involves the pyridyl group (see Figure 5a, and Figure A1 and Figure A2 of Appendix B). LUMO+1 is predominantly on the pyridyl group.

The presence of a 2-, 3- or 4-pyridyl group at the *meso*-position has a strong effect on the relative fluorescence quantum yields using rhodamine 6G as the reference (Φ_f_ = 0.86 in methanol) [40]. BODIPY **3Py** was found to be the most fluorescent (Φ_f_ = 0.43) and the **2Py** is the least (Φ_f_ = 0.04). The high fluorescence of **3Py** and the **4Py** in organic solvents has previously been observed [36,41,42]. On the other hand, the **2Py** displays much lower fluorescence compared with its regioisomers, as previously reported [35,43]. This is due to the closer proximity of the nitrogen atom to the BODIPY core in **2Py** relative to the **3Py** and **4Py** compounds. The 2-pyridyl group has a lower rotational barrier compared with the 3- and 4-pyridyl groups (see Table 2), suggesting that in the case of **2Py** more energy is released via non-radiative decay pathways. The *meso*-pyridyl rotation causes distortion of the BODIPY core that disrupts electron conjugation and could be the reason for the observed lower quantum yields for **2Py**. Another possible explanation is the difference in the S1 LUMO. In the case of **3Py** and the **4Py**, the S1 LUMO is almost entirely localized on BODIPY, whereas for **2Py**, the orbital is delocalized over BODIPY and pyridyl.

Methylation of the pyridyl group leads to a decrease in the molar extinction coefficients, as well as bathochromic shifts in the maximum absorption wavelengths of the cationic derivatives in acetonitrile, that was more pronounced for the **2catPy** (18 nm) relative to the **3catPy** (8 nm) and **4catPy** (8 nm). This tendency is also seen in the calculated maximum absorption wavelengths; all cationic BODIPYs show bathochromic shifts with that of **2catPy** (19 nm) being more pronounced than for **3catPy** (13 nm) and **4catPy** (12 nm), demonstrating similar shifts. For this series of methylated compounds, the HOMO is again almost entirely located on the BODIPY core but the LUMO and LUMO+1 switch compared with the non-methylated analogs. In this case, LUMO is mostly on the pyridinium group, whereas LUMO+1 is mostly on BODIPY core (see Figure 5 and Figure A1 and Figure A2 of Appendix B). This suggests that PeT occurs from the BODIPY core to the pyridinium group upon excitation, which quenches a high percentage of the fluorescence. The gaps between HOMO and LUMO+1 (LUMO_BODIPY_) are given in Table 2, as this is the transition corresponding to the fluorescence activity. The electron-withdrawing effect lowers both HOMO and LUMO_BODIPY_ with greater effect on LUMO_BODIPY_, thus decreasing the gap between these two orbitals, which is in agreement with the experimentally observed bathochromic shift.

The observed bathochromic shifts are similar in the case of **3catPy** and **4catPy** but different in the case of **2catPy**. This is likely due to the stronger electron-withdrawing effect of the 2-pyridinium group due to the proximity of the nitrogen atom to the BODIPY core. While in the case of the **3catPy** and **4catPy** the pyridinium nitrogen pulls electron density nearly equally from both sides of the BODIPY core, in the case of **2catPy** there is a direct pull of electron density from the nearby carbon-2 of the pyridyl group, which induces the observed enhanced bathochromic shift. This is in agreement with the observed similar orbital energies of HOMO and LUMO_BODIPY_ in the case of **3catPy** and **4catPy**, and lower orbital energies in the case **2catPy** (see Figure 5 and Figure A1 and Figure A2 of Appendix B). Furthermore, moderate redshifted emissions were observed for **2catPy** and **3catPy**, centered at 544 and 532 nm, respectively. In the case of **4catPy** a more pronounced redshift to 596 nm was observed, as we previously reported [33], likely due to the highly dipolar species formed upon excitation, with a pronounced charge-transfer character [44].

The fluorescence in acetonitrile of **3catPy** was drastically reduced to 0.006, as previously observed for **4catPy** [33], due to intramolecular charge transfer, since pyridinium is more easily reduced than pyridine. This has led to the exploration of photo- or chemical induced dealkylation of a 4-pyridinium quencher as a strategy to restore the fluorescence of the BODIPY chromophore [45,46,47]. On the other hand, the fluorescence of **2catPy** actually increased to 0.04 to 0.13 or 0.23 with I^−^ or PF_6_^−^ as the counter ion, respectively. The observed lower quantum yields for **3catPy** and **4catPy** compared with **3Py** and **4Py** are likely due to intramolecular charge transfer, since pyridinium is more easily reduced than pyridine. The energy difference between LUMO and LUMO+1 is only 0.30 eV in the case of **2catPy** and 0.79/0.89 eV in **3catPy/4catPy**, respectively (see Figure 5a, and Figure A1 and Figure A2 of Appendix B), suggesting easier charge-recombination and higher quantum yield in **2catPy**. Furthermore, **3catPy** and **4catPy** show higher probability for intersystem crossing compared with **2catPy**, as suggested by the lower S1→T energy differences (Table 2). The increase in fluorescence upon alkylation of **2Py** has been previously observed [35], and is in part due to the bulkiness of the alkyl group on the pyridinium nitrogen atom in close proximity to the BODIPY core. The larger counter ion (PF_6_^−^) induces the highest relative fluorescence in the **2catPy**, but this effect is not observed for the **3catPy** nor the **4catPy**. We hypothesize that in the case of the cationic 3- and 4-pyridinium-BODIPYs with iodide as counter ion, there is likely a close interaction of the BODIPYs with the iodide anion, since the pyridinium nitrogen in **3catPy** and **4catPy** is easily accessible. Therefore, the iodide anion in these cases is able to exert a heavy-atom effect that leads to ISC from the lowest singlet excited state (S_1_) to the triplet state of the cationic system [48]. On the other hand, in BODIPY **2catPy** the sterically hindered cationic nitrogen center prevents close interaction with the iodide ion.

In water, the fluorescence quantum yields decreased further due to the increase in solvent polarity which stabilizes the charge-transfer state leading to deactivation through ISC, as previously observed for **2catPy** [35] and **4catPy** [33]. However, the maximum absorption and emission observed for the **2catPy** and **3catPy** remained unchanged in acetonitrile, methanol, and water (see Table 1 and Appendix A).

The installation of the 2,6-dichloro substituents led to significant redshifted absorptions (25–27 nm), decreased molar absorptivities, red-shifted emissions (28–31 nm), and slight increased Stokes shifts of the neutral BODIPYs, as previously observed [33,49], due to the electron-withdrawing effect of the chlorine atoms. The chlorines at the 2,6-positions stabilize both the HOMO and LUMO, and this effect is more pronounced on the LUMO (see Figure 5 and Figure A1 and Figure A2 of Appendix B). As a result, the HOMO-LUMO gaps are smaller for the chlorinated derivatives, causing the observed redshifts of the absorption wavelengths. The performed calculations show an increase in the oscillator strengths upon introduction of the chlorine atoms. In addition, the experimental relative fluorescence quantum yields increased upon 2,6-dichlorination, likely due to reduction in nonradiative deactivation, as it was previously shown that the nonradiative rate constant significantly decreases with the introduction of chlorine groups [49]. Chlorination at the 2,6-positions also increased the fluorescence quantum yields of the BODIPYs due to the decrease in electronic charge density on the BODIPYs. Compound **2PyCl_2_** again showed relatively reduced fluorescence (Φ_f_ = 0.17) in comparison with **3PyCl_2_** and **4PyCl_2_** (both Φ_f_ = 0.58) due to its closer proximity of the nitrogen atom to the BODIPY core.

The corresponding methylated 2,6-dichlorinated analogues showed increase in fluorescence quantum yields in acetonitrile, particularly in the case of **3catPyCl_2_** which showed a 38-fold enhancement in fluorescence relative to **3catPy_._** In water, the absorption and emission wavelengths of the cationic derivatives did not change significantly, except in the case of **2catPyCl_2_**. Furthermore, the fluorescence quantum yields were partially enhanced upon introduction of the 2,6-chlorines, as we previously reported in the case of **4catPyCl_2_** [33]. A drastic increase was observed in the case of **2catPyCl_2_** (Φ_f_ = 0.24) which displayed over 200-fold increase in fluorescence relative to **2catPy,** whereas 100- and 10-fold enhancements were observed for **3catPyCl_2_** and **4catPyCl_2_**, respectively. The observed fluorescence enhancements are likely due to an alteration of the excited state lifetime resulting from the increase in the BODIPY oxidation potential of the chlorinated BODIPY derivatives in addition to the naturally higher oxidation potential of the 2-pyridinium BODIPY (see below), which increases the rate of charge recombination by magnifying the electron-hole electrostatic interaction. In agreement with these findings, up to 100-fold fluorescence enhancement was reported for a **2catPy** derivative bearing methylmercaptophenoxy groups in place of the fluorines on the boron atom, upon oxidation to the corresponding sulfoxides [50]. Furthermore, our results show that the fluorescence of **3catPyCl_2_** in water (Φ_f_ = 0.47) is the highest among the series of pyridinium-BODIPYs.

### 3.4. Electrochemical Properties

The electrochemical data for the pyridinium-BODIPYs **2catPy**, **3catPy** and **4catPy** with PF_6_^−^ as counterion are summarized in Table 3 and shown in Figure 6. Each pyridinium-BODIPY shows one quasi-reversible oxidation corresponding to the formation of the p-radical cation [51]. Due to the stronger electron-withdrawing effect of the 2-pyridinium group in **2catPy**, the oxidation potential is shifted positive by 100 mV compared with **4catPy** and **3catPy**. Each pyridinium-BODIPY shows reversible reductions that occur either through an apparent two-electron reduction or two separate single-electron reductions. **4catPy** undergoes a single two-electron reduction which corresponds to the reductions localized on the BODIPY core and the pyridinium group. It is likely that the initial reduction forms a radical which is more easily reduced than the initial BODIPY compound, resulting in the second electron transfer occurring almost immediately upon the formation of the neutral radical. These results are in agreement with previous published data [32]. Meanwhile, **2catPy** and **3catPy** show two sequential reversible one electron reductions, which suggest an increased stability of the radical formed upon the initial reduction (Figure 6a). Overall, **2catPy** and **3catPy** are more resistant to reduction to either the singly or doubly reduced state compared with **4catPy**. BODIPY **2catPy** undergoes its first reduction 30 mV more negative and its second reduction 230 mV more negative than the two-electron reduction seen in **4catPy**. BODIPY **3catPy** is more significantly affected as its first reduction appears 200 mV more negative and its second reduction appears 480 mV more negative than the two-electron reduction in **4catPy**. The first reduction for both **2catPy** and **3catPy** lies in the potential range usually seen for the reduction in the BODIPY core to its singly reduced state, whereas the second reduction may correspond to the reduction in the pyridinium group [51]. Thus, the influence of *meso*-pyridinium substituents on the BODIPY redox potentials are moderate (~170 mV) and consistent with the electron-withdrawing effects of the pyridinium groups. On the other hand, the introduction of the 2,6-dichloro groups induces larger effects, as discussed below.

Similar trends were seen with chlorinated versions **2catPyCl_2_**, **3catPyCl_2_**, and **4catPyCl_2_** (Figure 6b and Table 3). The 2,6-dichlorinated BODIPYs have the same number of redox events as their unchlorinated counterparts but the reduction and oxidation waves are shifted positive. The oxidation wave is irreversible for each at 100 mV/s but becomes more reversible at 1 V/s for **4catPyCl_2_**, and at 10 V/s for **2catPyCl_2_**. Compared with the corresponding unchlorinated BODIPYs at 100 mV/s, the anodic peak for each was shifted positive by 70 mV, 160 mV, and 180 mV for **2catPyCl_2_**, **3catPyCl_2_**, and **4catPyCl_2_**, respectively. As in **4catPy**, **4catPyCl_2_** undergoes a reversible two-electron reduction, which is shifted 180 mV positive compared with **4catPy**. Meanwhile **2catPyCl_2_** and **3catPyCl_2_** undergo two sequential one-electron reductions. The first reduction wave is reversible for both **2catPyCl_2_** and **3catPyCl_2_** and is shifted positive by 220 mV and 240 mV, respectively, when compared with their unchlorinated BODIPYs. The second, more negative reduction wave, is reversible in **2catPyCl_2_** but quasi-reversible in **3catPyCl_2_** and is shifted positive by 130 mV and 120 mV when compared with **2catPy** and **3catPy**, respectively. As expected, the introduction of the 2,6-dichloro groups has a larger influence on the electrochemical properties of BODIPYs than the *meso*-pyridinium substituents.

### 3.5. Cytotoxicity Properties

The dark and photocytoxicity of this series of BODIPYs was investigated in human carcinoma HEp2 cells and the results are summarized in Table 4 and shown in Appendix A. Very low cytotoxicity was observed for all the BODIPYs without 2,6-chlorine substitution (IC_50_ > 200 mM in the dark and >100 mM at 1.5 J/cm^2^). The neutral 2,6-dichloro derivatives showed slightly enhanced cytotoxicity, particularly the **3PyCl_2_** due to the introduction of the chlorine atoms. The most cytotoxic were found to be the cationic 2,6-dichloro derivatives **2catPyCl_2_** and **3catPyCl_2_**. These results are in agreement with previous observations showing that the introduction of heavier halogens at the 2,6-positions, such as iodides, drastically reduce the fluorescence and increase the phototoxicity of meso-(4-pyridinium)-BODIPYs toward human cancer cells [38] and pathogenic microorganisms [52,53,54].

## 4. Conclusions

The photophysical, electrochemical, and cytotoxic properties of *meso*-(2-pyridyl) and *meso*-(3-pyridyl)-BODIPYs, and their 2,6-dichloro and pyridinium derivatives were compared with those of their corresponding *meso*-(4-pyridyl) and *meso*-(4-pyridinium) analogs. Due to the stronger electron-withdrawing effect of the 2-pyridinium group in **2catPy**, its oxidation potential was shifted positive by 100 mV compared with **4catPy** and **3catPy**. BODIPYs **2catPy** and **3catPy** were found to be more resistant to reduction compared with **4catPy.** Unlike the presence of heavy halogens (Br, I) at the BODIPY 2,6-positions, which is known to enhance ISC, the presence of 2,6-chlorines was shown to enhance the relative fluorescence quantum yields of the BODIPYs in both organic solvents and water. Furthermore, the **2catPy** isomer showed the highest relative fluorescence quantum yield in organic solvents, which was further enhanced by the use of a bulky counter anion (PF_6_^−^). In water, over a 200-fold increase in fluorescence was observed for **2catPyCl_2_**, 100-fold increase for **3catPyCl_2_** and 10-fold increase for **4catPyCl_2_**. The fluorescence of **3catPyCl_2_** in water (F_f_ = 0.47) was found to be the highest among the series of BODIPYs. Our results show that the introduction of 2,6-chlorines in BODIPYs has a larger influence on their electrochemical properties than the meso-pyridyl substituents.

## Figures and Tables

**Figure 1 sensors-22-05121-f001:**
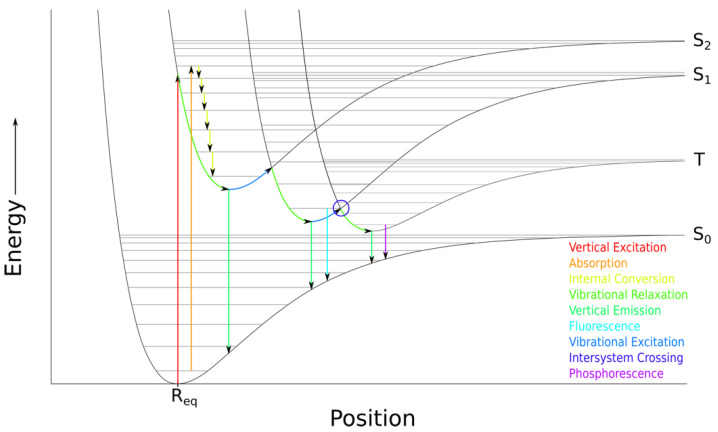
Simplified Franck–Condon energy well diagram.

**Figure 2 sensors-22-05121-f002:**
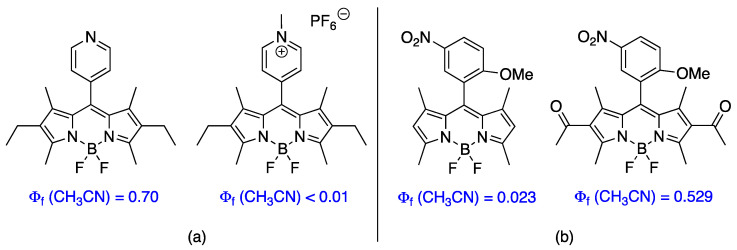
Previous work on modulating the fluorescent properties of BODIPY dyes: (**a**) effect of *N*-methylation of a *meso*-(4-pyridyl)-BODIPY [32], and (**b**) effect of introduction of acetyl groups at the 2,6-positions of a *meso*-aryl-BODIPY [34].

**Figure 3 sensors-22-05121-f003:**
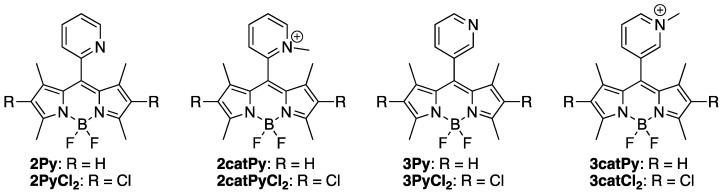
Structure of series of BODIPYs synthesized in this study.

**Figure 4 sensors-22-05121-f004:**
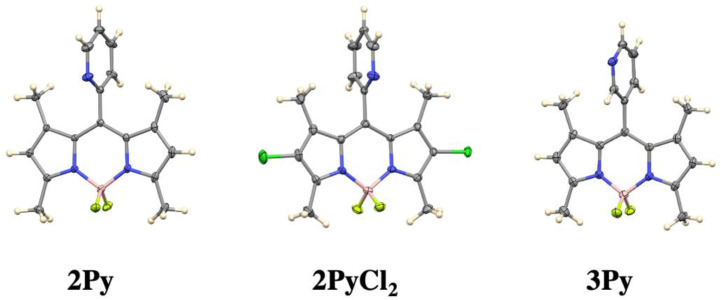
Crystal structures of **2Py**, **2PyCl_2_** and **3Py** BODIPYs with 50% ellipsoids.

**Figure 5 sensors-22-05121-f005:**
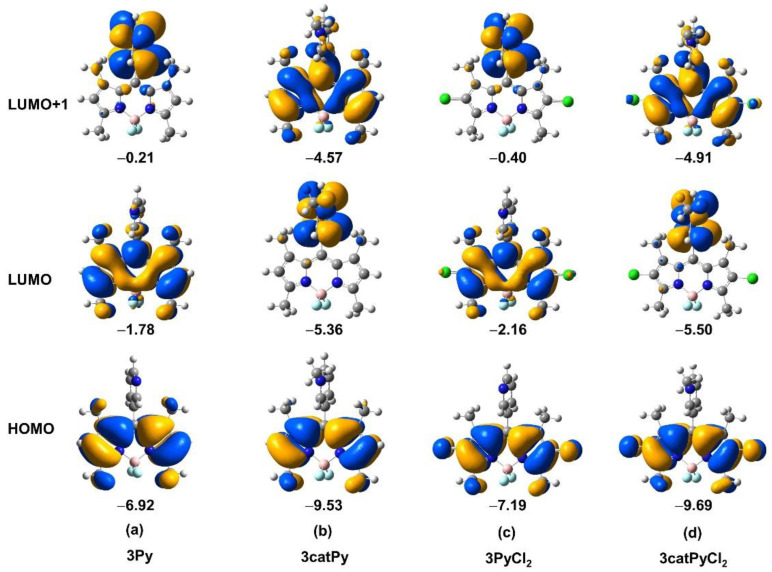
Frontier orbitals of (**a**) **3Py**, (**b**) **3catPy**, (**c**) **3PyCl_2_** and (**d**) **3catPyCl_2_** BODIPYs. Orbital energies in eV. The frontier orbitals for the entire series are given in Appendix B.

**Figure 6 sensors-22-05121-f006:**
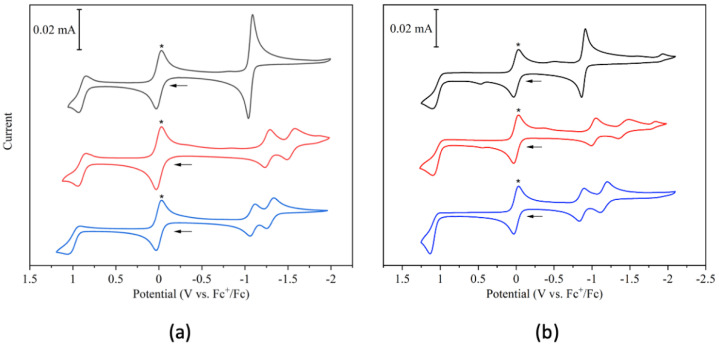
Cyclic voltammograms at 100 mV/s on platinum electrodes of 1 mM solutions of (**a**) **2catPy** (blue), **3catPy** (red), and **4catPy** (black) and (**b**) **2catPyCl_2_** (blue), **3catPyCl_2_** (red), and **4catPyCl_2_** (black), in degassed, anhydrous acetonitrile with 0.25 M NBu_4_PF_6_ supporting electrolyte. Arrows indicate the direction of the potential scan. The wave marked with an asterisk (*) at 0 V vs. Fc^+^/Fc corresponds to the internal Fc standard.

**Table 1 sensors-22-05121-t001:** Spectroscopic properties of BODIPYs in CH_3_CN and H_2_O and relative fluorescence quantum yields.

Solvent	BODIPY	λ_abs_ (nm)	λ_em_ (nm)	Stokes Shift (nm)	Φ_f_	ᵋ (M^−1^ cm ^−1^)
CH_3_CN	**2Py**	502	514	12	0.04 ^a^	87,500
	**3Py**	502	514	12	0.43 ^a^	92,900
	**4Py** [18]	501	515	14	0.31	72,100
	**2PyCl_2_**	530	545	15	0.17 ^a^	64,280
	**3PyCl_2_**	527	542	15	0.58 ^a^	57,540
	**4PyCl_2_** [18]	528	546	18	0.58	50,800
	**2catPy** (I^−^)	520	544	24	0.13 ^b^	65,800
	**2catPy** (PF_6_^−^)	520	546	26	0.23 ^b^	36,700
	**3catPy** (I^−^)	510	532	22	0.006 ^b^	65,460
	**3catPy** (PF_6_^−^)	510	529	19	0.006 ^b^	88,580
	**4catPy** (I^−^) [18]	509	596	87	0.019	26,000
	**4catPy** (PF_6_^−^)	509	603	94	0.010 ^b^	71,520
	**2catPyCl_2_** (I^−^)	530	547	17	0.17 ^b^	19,420
	**3catPyCl_2_** (I^−^)	540	560	20	0.23 ^b^	43,160
	**4catPyCl_2_** (I^−^) [18]	538	600	62	0.048	23,400
H_2_O	**2catPy** (I^−^)	520	542	22	0.001 ^b^	52,380
	**3catPy** (I^−^)	510	530	20	0.005 ^b^	37,320
	**4catPy** (I^−^) [18]	509	600	91	0.004	31,600
	**2catPyCl_2_** (I^−^)	555	577	22	0.24 ^b^	21,000
	**3catPyCl_2_** (I^−^)	540	564	24	0.47 ^b^	35,180
	**4catPyCl_2_** (I^−^)	540	605	65	0.038	11,640

^a^ Calculated using rhodamine 6G (Φ_f_ = 0.86) in methanol at λ_exc_ = 473 nm as the standard. ^b^ Calculated using Ru(bpy)Cl_2_.6H_2_O (Φ_f_ = 0.028) in H_2_O at λ_exc_ = 436 nm as the standard.

**Table 2 sensors-22-05121-t002:** Calculated spectroscopic properties, HOMO-LUMO_BODIPY_ gap, rotational barriers of the pyridyl or pyridinium groups with respect to the BODIPY core, and S1→T energy difference (cam-b3lyp/6-31+G(d,p) in gas phase). The LUMO_BODIPY_ is LUMO for the neutral species but LUMO+1 for the cationic species.

BODIPY	λ_abs_ (nm)	Oscillator Strength	HOMO-LUMO_BODIPY_ Gap (eV)	Rotation Barrier (kcal/mol)	S1-T Energy Difference (eV)
**2Py**	416	0.542	5.14	17.8	
**3Py**	416	0.542	5.14	20.7	
**4Py**	415	0.544	5.15	19.7	
**2catPy**	435	0.537	4.88	>35	0.057
**3catPy**	429	0.490	4.95	25.3	0.025
**4catPy**	427	0.550	4.98	25.1	0.012
**2PyCl_2_**	432	0.565	5.05	16.6	
**3PyCl_2_**	433	0.562	5.03	18.6	
**4PyCl_2_**	432	0.561	5.04	19.2	
**2catPyCl_2_**	460	0.558	4.72	-	
**3catPyCl_2_**	455	0.533	4.78	-	
**4catPyCl_2_**	451	0.567	4.82	-	

**Table 3 sensors-22-05121-t003:** Oxidation and reduction potentials for BODIPYs determined by cyclic voltammetry (in V vs. Fc^+^/Fc) at 100 mV/s in degassed and anhydrous acetonitrile with 0.25 M NBu_4_PF_6_ at analyte concentrations of 0.001 M.

BODIPY	$E_{1/2}^{ox}\;(V)$	$E_{1/2}^{red}\;(V)$
**2catPy**	+0.99 ^a^	−1.09; −1.29
**3catPy**	+0.89	−1.26; −1.54
**4catPy**	+0.89	−1.06 ^e^
**2catPyCl_2_**	+1.10 ^b^	−0.87; −1.16
**3catPyCl_2_**	+1.10 ^c^	−1.02; −1.42
**4catPyCl_2_**	+1.03 ^d^	−0.89 ^e^

^a^ Value estimated: this wave is more irreversible than the others. ^b^ Value at 10 V/s. ^c^ Value for oxidation peak potential only: this wave is irreversible. ^d^ Value at 1 V/s. ^e^ Two-electron reduction to doubly reduced state.

**Table 4 sensors-22-05121-t004:** Dark and photo (1.5 J/cm^2^) cytotoxicity of BODIPYs toward HEp2 cells.

BODIPY	Dark Toxicity (IC_50_, mM)	Phototoxicity (IC_50_, mM)
**2Py**	>200	>100
**3Py**	>200	>100
**4Py**	>200	>100
**2PyCl_2_**	>200	>100
**3PyCl_2_**	200	>100
**2catPy**	>200	>100
**3catPy**	>200	>100
**4catPy**	>200	>100
**2catPyCl_2_**	80	80
**3catPyCl_2_**	120	>100

## Data Availability

Not applicable.

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
