# Peer review of "A Comparison of the Photophysical, Electrochemical and Cytotoxic Properties of meso-(2-, 3- and 4-Pyridyl)-BODIPYs and Their Derivatives"

_sensors, 2022, doi:10.3390/s22145121_

Round 1

Reviewer 1 Report

In this work, Vicente and coworkers describe their comparison studies of the substitute effect on the electronic and photophysical properties of meso-pyridyl-BODIPYs. The influence of chlorine groups of the meso-pyridyl-BODIPYs and their corresponding methylated cationic pyridinium-BODIPYs were revealed through UV-Vis and fluorescence spectroscopy, CV and TD-DFT calculations. Structure-property relationships were established and interpreted in depth. The reviewer found this work very interesting. In particular, the study on the correlation between the PeT, ISC and fluorescence provides insightful and useful information. This study is well thought out. The manuscript is well written. The compounds are well characterized. The discussion is thorough and in depth. The reviewer recommends publication of this manuscript in Sensors. The review would like to make minor revision suggestion as listed below:

·       It is good to indicate the HOMO/LUMOs in Figure 4 and Figure A1 and Figure A2.

Author Response

Figures 4 (5 in the revised version), A1 and A2 were revised based on the reviewer comments, by introducing the LUMO+1/LUMO/HOMO labels

Reviewer 2 Report

In the present article authors report the synthesis of meso-(2-pyridyl) and meso-(3-pyridyl)-BODIPYs, and of their 2,6-dichloro and pyridinium derivatives (see Figure 66 2) and compare their photophysical, electrochemical and cytotoxic properties to different (already known BODIPYs) derivatives.

The manuscript is well written and experiments well conceived, probably in the introduction it would be advisable to better clarify merits and shortcomings of the present systems with respect to the previously reported ones to highlight their improved properties.

Author Response

We revised the Introduction, as suggested by the reviewer. The revise paragraph reads: "On the other hand, pyridinium groups are easy to install, and lead to cationic BODIPY derivatives with enhanced water-solubility, stability, and membrane permeability. In addition, meso-pyridyl-BODIPYs have been used as ligands in coordination chemistry, and as fluorescent pH sensors; therefore this type of functionalized BODIPY has found applications in bioimaging and biosensing."

Reviewer 3 Report

Vicente and co-workers synthesized two new BODIPY-derivatized fluorescent dyes with pyridinium groups that are water soluble, which showed high fluorescence quantum yield in water and good cytocompatibility. The authors performed systematic investigation on their crystal structures, spectrospcopic and electrochemical properties, and calculated their frontier orbitals, with reported BODIPY dyes as references. This is a solid work with interest for the field of molecular fluorophores. Thus I recommend its publication in Sensors. Here are some minor comments:

1. As the authors indicated the charge transfer in the pyridinum products, but I'm not sure what is the particular process of the charge transfer, as both BODIPY and pyridinum are electron-withdrawing?

2. It would be more clear to label HOMO, LUMO and LUMO+1 directly in Figure 4 and A1, A2;

It would be more reader friendly to move the key spectra from SI to the main article 

Author Response

We added a new Figure (Figure 1) to better explain the transitions that take place. We also revised the Introduction with the sentence: This process occurs by the excitation (S0→S2) and subsequent non-radiative decay from (S2→S1) moving the electron from HOMO→LUMO+1→LUMO as shown in Figure 1."

Figures 4 (5 in the revised version), A1 and A2 were revised as suggested by the reviewer. No additional Figures were moved from SM to the manuscript, as we already have six Figures in the manuscript plus two Figures in the Appendix.